# Prevalence of Cardiovascular–Kidney–Metabolic (CKM) Syndrome in Lithuanian Adults: Insights from a Nationwide Real-World Study Using Electronic Health Records

**DOI:** 10.3390/medicina61122106

**Published:** 2025-11-26

**Authors:** Gediminas Urbonas, Indrė Čeponienė, Inga Arūnė Bumblytė, Marius Miglinas, Lina Gatelytė, Živilė Steponkutė, Aušra Degutytė, Ingrida Grabauskytė, Džilda Veličkienė

**Affiliations:** 1Department of Family Medicine, Medical Academy, Lithuanian University of Health Sciences, Eiveniu 2, 50161 Kaunas, Lithuania; lina.gatelyte@stud.lsmu.lt (L.G.); zivile.steponkute@stud.lsmu.lt (Ž.S.); 2Department of Cardiology, Medical Academy, Lithuanian University of Health Sciences, Eiveniu 2, 50161 Kaunas, Lithuania; indre.ceponiene@lsmu.lt; 3Department of Nephrology, Lithuanian University of Health Sciences, Eiveniu 2, 50161 Kaunas, Lithuania; ingaarune.bumblyte@lsmu.lt; 4Clinic of Gastroenterology, Nephrourology and Surgery, Faculty of Medicine, Vilnius University, Santariškių 2, 03101 Vilnius, Lithuania; marius.miglinas@santa.lt; 5Patient Organisation “Gyvastis”, Žolyno g. 3, 10208 Vilnius, Lithuania; ausra.degutyte@gmail.com; 6Department of Physics, Mathematics and Biophysics, Lithuanian University of Health Sciences, Eivenių 4, 50103 Kaunas, Lithuania; ingrida.grabauskyte@lsmu.lt; 7Institute of Endocrinology, Lithuanian University of Health Sciences, Eiveniu 2, 50161 Kaunas, Lithuania; dzilda.velickiene@lsmu.lt

**Keywords:** cardiovascular disease, cardiovascular–kidney–metabolic syndrome, chronic kidney disease, diabetes, prediabetes, obesity

## Abstract

*Background and Objectives*: Cardiovascular–kidney–metabolic (CKM) syndrome reflects the interconnection between metabolic risk factors, chronic kidney disease (CKD), and cardiovascular disease (CVD). Despite increasing awareness, population-based data on CKM syndrome are limited, particularly in Europe. This study assessed the prevalence of CKM syndrome and the use of renal and cardiac biomarkers in Lithuania. *Materials and Methods*: Health records of 923,329 adults aged ≥40 years from the national Electronic Health Services and Cooperation Infrastructure Information System were analyzed. CKM-associated conditions (prediabetes/type 2 diabetes, obesity, CKD) and cardiovascular outcomes (atherosclerotic CVD, peripheral vascular disease, stroke, heart failure, atrial fibrillation) were identified. CKM stages were defined as stage 0 (no CKM conditions), stages 1–3 (at least one CKM condition), and stage 4 (at least one CVD diagnosis). The use of estimated glomerular filtration rate (eGFR), albumin-to-creatinine ratio (ACR) and N-terminal pro–B-type natriuretic peptide (NT-proBNP) testing was evaluated. *Results*: Overall, 34.8% of adults met criteria for stage 4 CKM syndrome, and 23.4% were classified as stage 1–3. Obesity (21.2%) and type 2 diabetes (17.2%) were the most common CKM-associated conditions. Heart failure (25.4%) and atrial fibrillation (14.0%) were the most common cardiovascular outcomes, with ≥2 CVD diagnoses present in 15.4% of patients. Among stage 1–3 patients, eGFR, ACR, and NT-proBNP were measured in 53.5%, 9.0%, and 4.9%, respectively. *Conclusions*: A third of Lithuanian adults aged ≥40 years had stage 4 CKM syndrome. The underuse of biomarker testing highlights missed opportunities for early detection. Broader implementation of biomarker testing and integrated care is warranted to slow progression of CKM syndrome and reduce cardiovascular risk.

## 1. Introduction

Cardiovascular diseases (CVDs) are the primary cause of morbidity, mortality, and premature death in Europe, accounting for 1.7 million deaths in the European Union in 2021 and imposing a major socioeconomic burden through disability and reduced productivity [1]. The situation is particularly severe in Eastern Europe, which has consistently reported the highest age-standardized CVD mortality rates worldwide, ranging from 215 to 553 deaths per 100,000 population in 2022 [2]. Alongside CVD, chronic kidney disease (CKD) is a growing global health challenge, affecting about 10% of adults worldwide [3,4]. In Europe, the prevalence of CKD shows marked variation, ranging from approximately 3.3% in Norway to 17.3% in Germany for CKD stages 1–5 [5]. CKD is now the third fastest-growing cause of death globally [3] and is projected to become the fifth leading cause by 2040 [4].

The close interrelation between cardiovascular and renal disorders is well-established, with dysfunction in one organ system often leading to functional impairment in the other through common pathophysiological mechanisms [6]. This bidirectional relationship gave rise to the concept of cardiorenal syndrome, which describes a spectrum of acute and chronic conditions involving the heart and kidneys [7]. Over time, the understanding of cardiorenal syndrome has evolved, highlighting the challenges in distinguishing the primary affected organ and capturing the full complexity of disease mechanisms [8,9]. Ultimately, it was suggested to substitute the term “cardiorenal syndrome” with “chronic cardiovascular and kidney disorder” to better capture the ongoing and systemic nature of the condition, rather than the late-stage organ failure that “syndrome” implies [10].

The American Heart Association (AHA) has recently introduced the broader concept of cardiovascular–kidney–metabolic (CKM) syndrome, defining it as a systemic condition marked by the interplay of metabolic risk factors, CKD, and the cardiovascular system [11]. This interaction results in dysfunction across multiple organs and a high incidence of cardiovascular outcomes. CKM syndrome encompasses individuals at risk for CVD due to metabolic risk factors, CKD, or both, as well as those already diagnosed with CVD that may be linked to or exacerbated by metabolic risk factors or CKD [10]. CKM syndrome typically arises from an excess of adipose tissue, its dysfunction, or a combination of both. Various pathological mechanisms associated with dysfunctional adipose tissue lead to insulin resistance and, eventually, hyperglycemia. Key processes such as inflammation, oxidative stress, insulin resistance, and vascular dysfunction lead to metabolic risk factors, the advancement of kidney disease, the enhancement of heart–kidney interactions, and the onset of CVD. Metabolic risk factors and CKD further increase the likelihood of CVD via various direct and indirect pathways [11].

To reflect the progressive pathophysiology of CKM and support its early detection, preventive and management strategies, the five stages of CKM syndrome are defined. Stage 0 indicates the absence of CKM risk factors; stage 1 involves excess or dysfunctional adiposity; stage 2 includes metabolic risk factors such as hypertension, diabetes, elevated triglycerides, or moderate-to-high-risk CKD; stage 3 reflects subclinical CVD or equivalent high-risk conditions; and stage 4 represents established CVD within the context of CKM syndrome [12].

It is estimated that 80–90% of adults in the United States (US) have at least one component of CKM syndrome [13,14] and about 13% are already in advanced stages 3 or 4 [13,15]. High prevalence of individuals with CKM syndrome was reported in South Korea [16,17], Taiwan [18], and China [19].

Despite growing awareness of CKM syndrome, data on its prevalence in large real-world populations remain scarce, and data from Europe are particularly lacking. To address this gap, we analyzed electronic health records of over 920,000 adults to assess the prevalence of CKM syndrome in Lithuania, with a specific focus on advanced CKM (stage 4). In addition, we also evaluated the use of renal and cardiac biomarkers, which may facilitate early detection of CKM-related dysfunction and support timely intervention.

## 2. Materials and Methods

This retrospective study analyzed national health data, extracted from the Electronic Health Services and Collaboration Infrastructure Information System (ESPBI IS) and provided by the Lithuanian State Data Agency (SDA). This digital platform is used by healthcare providers in to handle patient medical records. It includes clinical data such as diagnostic codes, laboratory and imaging results, referrals, prescribed and dispensed medications, and other medical details. Authorization for secondary use of health data was granted by the Lithuanian SDA (permission No. 33, issued on 31 March 2025).

Before analysis, all records retrieved from the ESPBI IS were completely anonymized, with all personal identifiers removed to ensure patient privacy. The study protocol received approval from the Kaunas Regional Biomedical Research Ethics Committee (permission No. BE-2-120). Since the study utilized anonymized data for secondary purposes, obtaining informed consent from individual patients was not necessary.

The study population consisted of all adult individuals aged 40 years and above who had data recorded in the ESPBI IS from 1 January 2024 to 31 December 2024. The SDA automatically extracted records from semi-structured entries in the observation section of the referral form (E027). This automated process utilized various pattern-matching rules, and the data provider estimates its accuracy to be greater than 95%. No exclusion criteria were applied.

The following CKM-associated conditions were assessed: obesity (International Classification of Diseases, 10th Revision [ICD-10] codes E65, E66, U78.1 or body mass index ≥ 30 kg/m^2^), prediabetes (E09, R73), type 2 diabetes (E11), and CKD (N18.3, N18.4, N18.5, U87.1). Cardiovascular diagnoses included atherosclerotic CVD (I21, I22, I25.2, Z95.5, Z95.1, Z95.8, I25.0, I25.1, I25.11), peripheral vascular disease (I70), stroke (I63), heart failure (HF; I11.0, I13.0, I13.2, I50), or atrial fibrillation (AF; code I48).

Patients were classified according to CKM syndrome stages. Stage 0 CKM was defined as absence of CKM-associated conditions and cardiovascular diagnoses. Stages 1–3 CKM were defined as the presence of at least one CKM-associated condition. Stage 4 CKM was defined as the presence of at least one cardiovascular diagnosis.

All available measurements of three biomarkers—estimated glomerular filtration rate (eGFR), urinary albumin-to-creatinine ratio (ACR) and N-terminal pro–B-type natriuretic peptide (NT-proBNP)—were included in the analysis irrespective of the test date. Only ACR values reported in mmol/L, which represented 87% of all ACR records, were included into analysis. For subgroup comparisons, ACR values were dichotomized as <3 mg/mmol and ≥3 mg/mmol. eGFR and NT-proBNP were analyzed as binary variables (test performed vs. not performed) only.

The analysis utilized multiple datasets extracted from the ESPBI IS, including files containing diagnostic codes, laboratory results for ACR, eGFR and NT-proBNP, as well as demographic information. These datasets were merged to create a single analytical database. Because data completeness varied among variables, the number of patients included in each analysis depended on the availability of relevant variables.

Descriptive statistics were applied to characterize the study population, including demographics, clinical parameters, and biomarker results. Qualitative variables were reported as counts and percentages, while quantitative variables as means (standard deviations (SD) and medians (minimum-maximum) are presented. Comparisons of proportions between groups were carried out using the chi-square test. Data transformations and descriptive statistics were performed on the SDA Reuse Platform, which is based on the original version of Palantir Foundry, utilizing its Contour tool for the data transformation processes. Other statistical analyses were conducted using IBM SPSS Statistics (version 29), with significance set at a two-sided *p* value of less than 0.05.

## 3. Results

According to official data from the Lithuanian SDA, there were 1,531,303 permanent residents aged ≥40 years as of 1 January 2024. Our study included 923,329 individuals meeting the eligibility criterion of having data recorded in the ESPBI IS during 2024. The mean age of the study population was 66.9 years, and 59.1% were men (Table 1).

Overall, 23.4% of the study population met criteria for stages 1–3 CKM syndrome and 34.8% met criteria for stage 4 CKM syndrome (Table 1). HF (25.4%) and AF (14.0%) were the most frequent cardiovascular diagnoses. Obesity was the most common CKM-associated condition (21.2%), followed by T2D (17.2%) and CKD (5.5%).

Biomarker testing across CKM syndrome stages, cardiovascular outcomes, and CKM-related conditions is summarized in Table 2. eGFR was measured in approximately half of the study population, while ACR and NT-proBNP tests were infrequent. NT-proBNP was most commonly assessed in patients with HF (22.4%) and those with multiple CVD diagnoses (24.5%). ACR testing was even less frequent, ranging from 17.2% among patients with CKD to only 5.4% among those with stroke. Among individuals with stage 1–3 CKM syndrome, ACR results were available for only 9.0%, and NT-proBNP results were available for 4.9% (Table 2).

Among 43,349 patients with both eGFR and ACR results available, elevated ACR was often observed even when eGFR indicated normal or mildly reduced kidney function (≥60 mL/min/1.73m^2^). The proportion of patients with ACR ≥3 mg/mmol was significantly higher than that with normal ACR values across CKM syndrome stages, CKM-associated conditions, and most cardiovascular outcomes, except for peripheral artery disease and stroke (Table 3).

## 4. Discussion

In this large population-based study including more than 920,000 adults aged 40 years and older, we assessed the prevalence of CKM syndrome and evaluated the current use of renal and cardiac biomarkers. To our knowledge, these are the first nationally representative data describing the epidemiology of CKM syndrome in the Lithuanian population.

We found that 34.8% of our study population met criteria for stage 4 CKM syndrome, defined as the presence of at least one CVD. Direct comparison with other studies is limited by differences in staging definitions and age of study populations. Our analysis included adults aged ≥40 years, while studies based on US and Korean NHANEs data included individuals aged ≥20 years. In a US study, analysis of National Health and Nutrition Examination Survey (NHANES) 2011–2020 data showed a 9.2% prevalence of stage 4 based on self-reported CVD (coronary heart disease, angina, heart attack, HF, and stroke), although atrial fibrillation and peripheral artery disease were not captured [13]. Earlier US NHANES data (1988–2018) indicated stage 4 prevalence ranging from 4.2% in females and 6.3% in males in 1988–1994 to 6.0% in females and 7.2% in males in 2009–2018, using a definition based on coronary heart disease, HF, or stroke among individuals with excess/dysfunctional adiposity, other CKM risk factors, or CKD [20]. Another US study using data of the Behavioral Risk Factor Surveillance System, a nationally representative survey by the Centers for Disease Control and Prevention, reported a 6.7% prevalence of stage 4 CKM syndrome in adults aged ≥18 years, defined as the presence of myocardial infarction, angina, coronary heart disease, or stroke in addition to stage 1–3 criteria [21]. Two South Korean studies using Korean NHANES data reported lower prevalence of stage 4: 2.8% in the study by Yim et al. [17], which defined stage 4 as clinical CVD in individuals with excess/dysfunctional adiposity, other metabolic risk factors, or CKD, and 3.9% in the study by Hong et al. [16], which defined stage 3–4 as very high-risk CKD or clinical CVD (stroke, angina pectoris, myocardial infarction). Similarly, in Taiwan, Tsai et al. (2025) found a 3.8% prevalence using a definition based on self-reported medication usage or the pursuit of medical care for coronary heart disease, HF, or stroke or presence of CKD [18]. In a Chinese cohort of healthcare workers, stage 4 prevalence was 5.1% among men and 4.3% among women, although diagnostic criteria were not reported [19].

Individuals with more advanced stages of CKM face substantially higher risk of mortality and reduced life expectancy. Lei et al. [22] demonstrated that, compared with stage 0, individuals at stage 4 had a four-fold increased risk of all-cause mortality and lost 15.5 years of life at the age of 50 years. Similarly, Claudel et al. [23] reported a 9.6% absolute difference in cardiovascular mortality over 15 years between stage 4 and stage 0, with the survival difference of 8.1 months. Although our study is cross-sectional and cannot provide direct prognostic data, the fact that more than one-third of Lithuanian adults aged ≥40 years already fulfil stage 4 CKM syndrome criteria suggests substantial population-level implications for premature cardiovascular death and reduced life expectancy.

Given the substantial prevalence of CKM syndrome and severe consequences of its advanced stages, there is a clear need for reliable diagnostics tests and biomarkers that facilitate early identification and management of CKM-related dysfunction [24,25]. Current research is evaluating the potential benefits of population-wide screening and exploring how biomarker profiles may enable earlier recognition of subclinical organ damage, enhance diagnostic accuracy and risk stratification, and support individualized therapeutic strategies that could slow CKM progression and improve long-term outcomes [25,26]. Biomarkers under investigation in the context of CKM syndrome include ACR, NT-proBNP, high-sensitivity cardiac troponin, and lipoprotein A, as they enable early detection of subclinical organ injury [25,26].

Urinary ACR testing represents a cornerstone for the detection of albuminuria, which has been associated with increased cardiovascular mortality among individuals with diabetes, hypertension, CKD, or HF, as well as adults with multiple CVD risk factors. Consequently, ACR has been proposed as a promising, comprehensive biomarker for cardiovascular, kidney, and metabolic disorders, offering the potential to identify and monitor CKM syndrome progression earlier than waiting for a decline in eGFR [27].

Natriuretic peptides, particularly NT-proBNP, enable the detection of subclinical cardiac dysfunction [26] and provide valuable prognostic information on mortality risk in patients with diabetes, kidney disease, or CVD, while further research is needed to investigate its role in optimizing cardiorenal therapies [25].

In our study, among individuals with stage 1–3 CKM syndrome, eGFR was measured in 53.5% of patients, whereas ACR was assessed in only 9%. These findings are consistent with the previous reports of suboptimal utilization of these biomarkers, particularly ACR, in clinical practice [28,29,30,31], despite clear guideline recommendations to incorporate eGFR and ACR testing into routine cardiovascular and renal risk assessment [32,33,34,35].

Our findings underscore the considerable burden of advanced CKM syndrome in the general population and highlight key opportunities for intervention. Early CKM is thought to originate from excess or dysfunctional adiposity, which drives the majority of associated risk factors [11,36]. In our study, 21% of adults aged ≥40 years were obese, representing a population at increased risk of CKM progression and highlighting a window for timely preventive strategies. Evidence suggests that weight reduction strategies may help stabilize patients in the early stages of CKM syndrome and prevent further disease advancement [23]. In addition, we found that among patients with preserved eGFR, elevated ACR was frequently observed across CKM-associated conditions and cardiovascular outcomes, confirming its value for early identification of high-risk individuals.

Although early CKM stages, particularly stage 1, clearly represent the main target for prevention of and reductions in CVD and CKD [12], meaningful benefits can still be achieved in later stages. Interventions at stage 3 may modify disease progression, as evidenced by a 6.5% lower cardiovascular mortality risk compared with stage 4 [23]. Optimal use of pharmacological agents addressing interrelated cardiovascular, renal, and metabolic risks such as sodium-glucose co-transporter-2 inhibitors, mineralocorticoid receptor antagonists, glucagon-like peptide-1 receptor agonists, and renin–angiotensin–aldosterone system inhibitors is essential for individuals showing signs of cardiovascular or kidney impairment [27].

CKM syndrome is viewed as a dynamic continuum, where early metabolic disturbances can progressively evolve into CKD and CVD if left unaddressed. This highlights the importance of implementing early, coordinated, and comprehensive interventions aimed at stopping disease progression before permanent organ damage develops [26]. Effective CKM management requires a holistic, multidisciplinary approach, ensuring that cardiologists, nephrologists, diabetologists, and general practitioners collaborate to address shared risk factors and tailor interventions to disease stage and severity [26].

The results of our study imply that strengthening metabolic health, obesity- and kidney-centered strategies are particularly critical, given the high overlap between CKD and cardiovascular outcomes and the crucial role of metabolic risk factors. Expanding the use of eGFR and ACR testing, implementing risk prediction tools such as the Kidney Failure Risk Equations, and adopting evidence-based pharmacological therapies could enhance early detection and intervention. National initiatives focused on CKM screening, standardized biomarker protocols, longitudinal monitoring of kidney function, and integrated, multidisciplinary care models would represent key steps toward reducing CKM syndrome progression and improving long-term outcomes. In clinical practice, annual (or more frequent) measurement of eGFR and urinary ACR should become routine in patients with obesity, prediabetes or diabetes, hypertension, or any cardiovascular diagnosis, representing the vast majority of individuals with CKM stages 1–4. ACR is particularly important because it detects renal damage and strongly predicts cardiovascular events years before eGFR decline becomes apparent. NT-proBNP should be considered in patients with heart failure, atrial fibrillation, multiple cardiovascular diagnoses, or when subclinical cardiac strain is suspected. The recently proposed ABCDE framework (albuminuria, blood pressure, cholesterol, diabetes, eGFR) [37], a simple yet comprehensive approach aligned with cardiovascular risk stratification recommended by the 2021 European Society of Cardiology Prevention Guidelines, should be incorporated into national healthcare strategies to facilitate systematic cardiovascular and renal risk assessment.

The strengths of this study include its large sample representing the Lithuanian adult population and the comprehensive assessment of both cardiovascular outcomes and CKM-associated conditions using real-world data.

We acknowledge several limitations of our study. First, the study’s focus on adults aged ≥40 years with recorded ESPBI IS data during 2024 may have introduced selection bias, excluding younger at-risk individuals or those without recent healthcare interactions, and thus not fully represent the broader Lithuanian population. Second, due to the inherent constraints of data derived from electronic health care records, the classification of CKM stages could not fully align with the official AHA criteria. In particular, incomplete information for certain key variables precluded the differentiation of intermediate CKM stages; therefore, stages 1–3 were grouped together for analysis. In addition, the overall prevalence of stage 1–3 CKM syndrome in our study was likely underestimated due to this reason. It should be noted, however, that some methodological adaptations have been applied in other population-based studies, reflecting a common limitation in real-world CKM research. Third, reliance on ICD-10 diagnostic codes for identifying CKM-associated conditions and CVD outcomes may have introduced misclassification bias, as coding accuracy can vary by clinician, institution, or documentation practices, potentially leading to under-reporting of conditions such as obesity or CKD. The observed CKD prevalence of 5.5% is lower than expected based on European epidemiological data (prevalence of CKD stages 1–5 up to 17% in adults aged 20–74 years [5]), suggesting that early CKD might have been missed when relying solely on diagnostic codes. Most likely, underutilization of renal function markers in routine care may have contributed to this underestimation. Several factors may explain the limited use of eGFR and ACR testing in primary care: family physicians may have insufficient awareness of CKD risk factors and guideline-recommended screening practices; serum urea, rather than creatinine, is still commonly ordered despite its limited diagnostic value; and in some laboratories, eGFR is not automatically calculated from serum creatinine, leading to further missed opportunities for early detection. Together, these gaps likely contributed to under-recognition of CKD and, consequently, advanced CKM syndrome. Finally, all available biomarker measurements were included in the analysis irrespective of the test date. Consequently, the actual utilization of these biomarkers may be even lower than reported.

## 5. Conclusions

This study provides the first nationally representative data on CKM syndrome in Lithuania. Among adults aged 40 years and older, approximately a third of individuals met the criteria for stage 4 CKM syndrome, reflecting a substantial burden of concurrent metabolic, renal, and cardiovascular disorders. The limited use of ACR and NT-proBNP testing highlights missed opportunities for early detection of CKM syndrome. Broader, guideline-directed implementation of annual eGFR and especially ACR testing, supplemented by selective use of NT-proBNP testing in high-risk groups, represents an immediate and cost-effective strategy to identify individuals with subclinical cardiorenal damage, improve risk stratification, and initiate or intensify reno- and cardioprotective therapies at a still-reversible stage of CKM syndrome. Strengthening integrated, multidisciplinary care that links primary care, cardiology, endocrinology, and nephrology is essential to ensure timely diagnosis, coordinated management, and reductions in CKM-related morbidity and mortality at the population level.

## Figures and Tables

**Table 1 medicina-61-02106-t001:** Demographic and clinical characteristics of study population.

Characteristics	n = 923,329
Men, n (%)	546,103 (59.1)
Women, n (%)	377,226 (40.9)
Age (years), mean (SD)	66.9 (12.3)
Age (years), median (min-max)	66 (43–110)
Body mass index (kg/m^2^), mean (SD)	29.5 (5.9)
Body mass index (kg/m^2^), median (min-max)	28.9 (15–50)
** *CKM syndrome stages* **	
Stage 0	386,320 (41.8)
Stages 1–3	215,707 (23.4)
Stage 4	321,302 (34.8)
** *Cardiovascular outcomes of CKM syndrome* **	
Atherosclerotic CVD, n (%)	83,010 (9.0)
Peripheral vascular disease, n (%)	34,728 (3.8)
Stroke, n (%)	19,177 (2.1)
Heart failure, n (%)	234,383 (25.4)
Atrial fibrillation, n (%)	129,170 (14.0)
≥2 of above diagnoses, n (%)	141,794 (15.4)
** *CKM-associated conditions* **	
Prediabetes, n (%)	56,634 (6.1)
Type 2 diabetes, n (%)	158,897 (17.2)
Chronic kidney disease, n (%)	51,154 (5.5)
Obesity, n (%)	195,343 (21.2)

CKM, cardiovascular–kidney–metabolic; CVD, cardiovascular disease; SD, standard deviation.

**Table 2 medicina-61-02106-t002:** Proportion of patients with CKM-associated conditions and cardiovascular outcomes who underwent NT-proBNP, eGFR and ACR testing.

	eGFR Test Done	ACR Test Done	NT-proBNP Test Done
** *CKM syndrome stages* **			
Stage 0, n (%)	145,575 (37.7)	9851 (2.5)	14,891 (3.9)
Stages 1–3, n (%)	115,375 (53.5)	19,426 (9.0)	10,628 (4.9)
Stage 4, n (%)	161,846 (50.4)	21,382 (6.7)	60,881 (18.9)
** *Cardiovascular outcomes of CKM syndrome* **			
Atherosclerotic CVD, n (%)	45,565 (54.9)	6478 (7.8)	18,199 (21.9)
Peripheral vascular disease, n (%)	19,057 (54.9)	2861 (8.2)	5179 (14.9)
Stroke, n (%)	9110 (47.5)	1029 (5.4)	2928 (15.3)
Heart failure, n (%)	120,330 (51.3)	16,621 (7.1)	52,462 (22.4)
Atrial fibrillation, n (%)	68,591 (53.1)	8464 (6.6)	27,962 (21.6)
≥2 of above diagnoses, n (%)	77,774 (54.8)	10,556 (7.4)	34,730 (24.5)
** *CKM-related conditions* **			
Prediabetes or Type 2 diabetes, n (%)	115,972 (55.8)	27,700 (13.3)	24,150 (11.6)
Chronic kidney disease, n (%)	39,056 (76.3)	8807 (17.2)	11,953 (23.4)
Obesity, n (%)	113,365 (58.0)	18,570 (9.5)	24,575 (12.6)

Percentages were calculated using number of patients with a condition or diagnosis as a denominator. ACR, albumin-to-creatinine ratio; CKM, cardiovascular–kidney–metabolic; CVD, cardiovascular disease; eGFR, estimated glomerular filtration rate; NT-proBNP, N-terminal pro–B-type natriuretic peptide.

**Table 3 medicina-61-02106-t003:** ACR results among patients with eGFR ≥60 mL/min/1.73m^2^.

	ACR Results	
	≥3 mg/mmol	<3 mg/mmol
** *CKM stages* **		
Stage 0		
N	7554	598
n (%)	5451 (72.2) *	386 (64.5)
Stages 1–3		
N	13,465	3211
n (%)	9527 (70.6) *	2169 (67.5)
Stage 4		
N	15,586	2935
n (%)	9371 (60.1) *	1595 (54.3)
** *Cardiovascular outcomes of CKM syndrome* **		
Atherosclerotic CVD		
N	4756	937
n (%)	2841 (59.7) ***	513 (54.7)
Peripheral vascular disease		
N	2117	467
n (%)	1233 (58.2)	252 (54.0)
Stroke		
N	756	139
n (%)	419 (55.4)	85 (61.2)
Heart failure		
N	12,136	2,279
n (%)	7143 (58.9) *	1196 (52.5)
Atrial fibrillation		
N	6,291	1,137
n (%)	3479 (55.3) *	517 (45.5)
≥2 of above CVD diagnoses		
N	7,814	1,489
n (%)	4364 (55.8) *	720 (48.4)
** *CKM-associated conditions* **		
Prediabetes or Type 2 diabetes		
N	18,873	4913
n (%)	13,162 (69.7) *	3256 (66.3)
Chronic kidney disease		
N	6973	1436
n (%)	2388 (34.2) **	431 (30.0)
Obesity		
N	13,396	2738
n (%)	9231 (68.9) *	1756 (64.1)

N denotes number of patients with eGFR ≥ 60 mL/min/1.73m^2^ (denominator). ACR, albumin-to-creatinine ratio; CKM, cardiovascular–kidney–metabolic; CVD, cardiovascular disease. * *p* < 0.001; ** *p* = 0.002; *** *p* = 0.005 (chi-square test).

## Data Availability

Data available on request due to privacy and legal reasons.

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
