# Peer review of "Prevalence of Cardiovascular–Kidney–Metabolic (CKM) Syndrome in Lithuanian Adults: Insights from a Nationwide Real-World Study Using Electronic Health Records"

_medicina, 2025, doi:10.3390/medicina61122106_

Round 1
Reviewer 1 Report
Comments and Suggestions for Authors
This is a well-done study which involved the use of a very large database of Lithuania. This study demonstrates that many patients do not have an adequate laboratory evaluation. In particular, they do not have albumin/creatinine ratios determined. This gives important information about renal function and probably should be concluded an evaluation of most patients at some time during their care. You should probably emphasize the laboratory tests which need more attention during the long long-term care of patients.
Author Response
Dear Reviewer,
We have substantially expanded the clinical implications both in the Discussion and Conclusions to explicitly highlight which laboratory tests deserve priority in the long-term management of patients at risk of or with established CKM syndrome. The following points have been emphasized (new text in lines 293–300):
„In clinical practice, annual (or more frequent) measurement of eGFR and urinary ACR should become routine in patients with obesity, prediabetes or diabetes, hypertension, or any cardiovascular diagnosis, representing the vast majority of individuals with CKM stages 1–4. ACR is particularly important because it detects renal damage and strongly predicts cardiovascular events years before eGFR decline becomes apparent. NT-proBNP should be considered in patients with heart failure, atrial fibrillation, mul-tiple cardiovascular diagnoses, or when subclinical cardiac strain is suspected.“
We have also added a specific recommendation in the Conclusions:
“ Broader, guideline-directed implementation of annual eGFR and especially ACR testing, supplemented by selective use of NT-proBNP testing in high-risk groups, represents an immediate and cost-effective strategy to identify individuals with subclinical cardi-orenal damage, improve risk stratification, and initiate or intensify reno- and cardio-protective therapies at a still reversible stage of CKM syndrome. Strengthening integrated, multidisciplinary care that links primary care, cardiology, endocrinology, and nephrology is essential to ensure timely diagnosis, coordinated management, and re-duction of CKM-related morbidity and mortality at the population level.”
Reviewer 2 Report
Comments and Suggestions for Authors
This manuscript analyzes the prevalence of CKM syndrome using national health data. The authors used a robust dataset.
My comments are:
CKD is one CKM-associated condition, but eGFR was not estimated in all patients (percentage of patients with known ACR was even lower), so the prevalence of CKD seems low. This may have an impact on the results. Authors should comment on this.
What were the consequences of CKM syndrome? Any information about mortality/prognosis?
Thank you
Author Response
Dear Reviewer,
Thank you for the constructive feedback that has significantly improved the article.
We believe these revisions directly address the Editor’s concerns and increase the clinical relevance of our article.
All changes are tracked in the revised manuscript.
Response for the comment 1:
We fully agree that the relatively low reported prevalence of CKD (5.5 %) in our study is largely explained by the incomplete availability of kidney function data in routine clinical practice and by the reliance on ICD-10 diagnostic codes rather than universal laboratory-based screening.
- eGFR was available in only ~50 % of the entire cohort (53.5 % in stages 1–3 CKM and 50.4 % in stage 4 CKM).
- ACR was measured in only 9.0 % of patients in stages 1–3 CKM and in 6.7 % of stage 4 patients, and even among patients with a registered CKD diagnosis only 17.2 % had an ACR result.
Because of this undertesting, a substantial proportion of CKD (especially early-stage albuminuric CKD with preserved eGFR) remained undetected and was therefore not coded. This is further supported by our finding that, among the 43,349 patients who had both eGFR ≥60 mL/min/1.73 m² and ACR measured, albuminuria (ACR ≥3 mg/mmol) was present in 55–72 % depending on CKM stage and co-existing conditions (Table 3). This clearly demonstrates that when the tests are actually performed, early kidney damage is very common even in patients with “normal” eGFR.
Thus, the true prevalence of CKD in the Lithuanian population aged ≥40 years is very likely significantly higher than the 5.5 % identified by diagnostic codes alone. Consequently, the real burden of CKM syndrome (especially stages 2–4) is also underestimated in our analysis.
We have expanded now added a dedicated paragraph in the Discussion section (lines 322–333) with the following text:
“ The observed CKD prevalence of 5.5 % is lower than expected based on European epi-demiological data (prevalence of CKD stages 1–5 up to 17% in adults aged 20-74 years [5]), suggesting that early CKD might have been missed when relying solely on diag-nostic codes. Most likely, underutilization of renal function markers in routine care may have contributed to this underestimation. Several factors may explain the limited use of eGFR and ACR testing in primary care: family physicians may have insufficient awareness of CKD risk factors and guideline-recommended screening practices; serum urea, rather than creatinine, is still commonly ordered despite its limited diagnostic value; and in some laboratories, eGFR is not automatically calculated from serum cre-atinine, leading to further missed opportunities for early detection. Together, these gaps likely contributed to under-recognition of CKD and, consequently, advanced CKM syndrome..”
Response for the comment 2:
Because the current dataset is cross-sectional and covers only year 2024 registrations, direct mortality or prognostic outcomes are not yet available. However, we have strengthened the Discussion section by cit explicitly linking the most recent and robust published evidence on CKM prognosis (Lei et al., Claudel et al.) to our population (new text in lines 228–231):
“ Although our study is cross-sectional and cannot provide direct prognostic data, the fact that more than one-third of Lithuanian adults aged ≥40 years already fulfil stage 4 CKM syndrome criteria suggests substantial population-level implications for premature cardiovascular death and reduced life expectancy”